# COVID-19 Complications: Oxidative Stress, Inflammation, and Mitochondrial and Endothelial Dysfunction

**DOI:** 10.3390/ijms241914876

**Published:** 2023-10-04

**Authors:** Ekaterina Georgieva, Julian Ananiev, Yovcho Yovchev, Georgi Arabadzhiev, Hristo Abrashev, Despina Abrasheva, Vasil Atanasov, Rositsa Kostandieva, Mitko Mitev, Kamelia Petkova-Parlapanska, Yanka Karamalakova, Iliana Koleva-Korkelia, Vanya Tsoneva, Galina Nikolova

**Affiliations:** 1Department of General and Clinical Pathology, Forensic Medicine, Deontology and Dermatovenerology, Medical Faculty, Trakia University, 11 Armeiska Str., 6000 Stara Zagora, Bulgaria; julian.r.ananiev@trakia-uni.bg; 2Department of Surgery and Anesthesiology, University Hospital “Prof. Dr. St. Kirkovich”, 6000 Stara Zagora, Bulgaria; yovcho.yovchev@trakia-uni.bg (Y.Y.); georgi.arabadzhiev@trakia-uni.bg (G.A.); 3Department of Vascular Surgery, Medical Faculty, Trakia University, 11 Armeiska Str., 6000 Stara Zagora, Bulgaria; hristo.abrashev@trakia-uni.bg; 4II Department of Internal Medicine Therapy: Cardiology, Rheumatology, Hematology and Gastroenterology, Medical Faculty, Trakia University, 6000 Stara Zagora, Bulgaria; despina.abrasheva@trakia-uni.bg; 5Forensic Toxicology Laboratory, Military Medical Academy, 3 G. Sofiiski, 1606 Sofia, Bulgaria; vatanasov@chem.uni-sofia.bg (V.A.); toxilab@vma.bg (R.K.); 6Department of Diagnostic Imaging, University Hospital “Prof. Dr. St. Kirkovich”, 6000 Stara Zagora, Bulgaria; mitko.mitev@trakia-uni.bg; 7Department of Medical Chemistry and Biochemistry, Medical Faculty, Trakia University, 11 Armeiska Str., 6000 Stara Zagora, Bulgaria; kamelia.parlapanska@trakia-uni.bg (K.P.-P.); yanka.karamalakova@trakia-uni.bg (Y.K.); 8Department of Obstetrics and Gynaecology Clinic, University Hospital “Prof. St. Kirkovich”, 6000 Stara Zagora, Bulgaria; iliana.koleva@trakia-uni.bg; 9Department of Propaedeutics of Internal Medicine and Clinical Laboratory, Medical Faculty, Trakia University, 11 Armeiska Str., 6000 Stara Zagora, Bulgaria; v.tsoneva.ivanova@trakia-uni.bg

**Keywords:** COVID-19, ROS, RNS, endothelial damage, mitochondrial dysfunction, oxidative stress, inflammation

## Abstract

SARS-CoV-2 infection, discovered and isolated in Wuhan City, Hubei Province, China, causes acute atypical respiratory symptoms and has led to profound changes in our lives. COVID-19 is characterized by a wide range of complications, which include pulmonary embolism, thromboembolism and arterial clot formation, arrhythmias, cardiomyopathy, multiorgan failure, and more. The disease has caused a worldwide pandemic, and despite various measures such as social distancing, various preventive strategies, and therapeutic approaches, and the creation of vaccines, the novel coronavirus infection (COVID-19) still hides many mysteries for the scientific community. Oxidative stress has been suggested to play an essential role in the pathogenesis of COVID-19, and determining free radical levels in patients with coronavirus infection may provide an insight into disease severity. The generation of abnormal levels of oxidants under a COVID-19-induced cytokine storm causes the irreversible oxidation of a wide range of macromolecules and subsequent damage to cells, tissues, and organs. Clinical studies have shown that oxidative stress initiates endothelial damage, which increases the risk of complications in COVID-19 and post-COVID-19 or long-COVID-19 cases. This review describes the role of oxidative stress and free radicals in the mediation of COVID-19-induced mitochondrial and endothelial dysfunction.

## 1. Introduction

Increasing scientific evidence confirms that the combination of ROS overproduction, oxidative stress (OS), and hyperinflammation during SARS-CoV-2 can cause endothelial layer damage, which eventually leads to endothelial dysfunction [1]. SARS-CoV-2-infection-initiated endothelial dysfunction can promote chronic inflammation, thrombosis, atherosclerosis, and lung injury. Various clinical, laboratory, and immunohistochemistry studies reveal that COVID-19 causes more than just viral pneumonia. As a result of the lethal inflammatory cytokine storm of COVID-19, apoptosis, inflammation, and endothelial dysfunction are observed, and are thought to result from mitochondrial dysregulation and impaired redox signaling. The generation of a large amount of mitochondrial reactive oxygen species (mtROS) and their excess causes OS, which can promote inflammation and cause chronic endothelial dysfunction [2].

Endothelial dysfunction increases blood clotting and microthrombi formation, contributing to severe complications of COVID-19. SARS-CoV-2 induces a prothrombotic state, manifested mostly by microthrombotic events known as immunothrombosis. It involves cooperation between the immune and coagulation systems and aims to initially orchestrate virus entry and block it [3]. Various pathways are involved in endothelial damage, including the overproduction of proinflammatory cytokines, hypoxia, and exocytosis, leading to the initiation of micro- and macrovascular thrombosis. The shedding of thrombomodulin from the endothelial cell surface further promotes a procoagulant and proinflammatory local environment, affecting pulmonary microcirculation [4]. Damage to the epithelial lining of the airways through the release of cytokines and chemokines from immune cells is a hallmark of COVID-19. The underlying mechanisms of endothelial damage in COVID-19 are still being studied, but viral replication, the immune response, inflammatory mediators, and mitochondrial dysfunction are believed to be major contributors. Endothelial damage is a potential complication of COVID-19 and leads to impaired blood flow and increased risk of thrombosis and acute respiratory distress syndrome (ARDS). The severity of epithelial damage in COVID-19 can vary widely, with factors such as advanced age, chronic disease, and compromised immune status influencing the degree of epithelial involvement [5].

This review discusses the relationship between SARS-CoV-2 (the virus responsible for COVID-19), ROS, and oxidative stress in mediating mitochondrial dysfunction, endothelial damage, and inflammation. It also touches upon the potential long-term consequences of high levels of oxidants and chronic redox imbalance in the body, often referred to as post-COVID-19 complications.

## 2. Literature Search, Methodology, Inclusion and Exclusion Criteria

Database and search strategy the number of original articles identified that reported the association between redox imbalance, ROS and RNS, and endothelial and mitochondrial dysfunction in different acute conditions and chronic diseases was 2073. The databases PubMed, PMC Europe, Scopus, Web of Science, and Google Scholar were used. Data analysis and preparation in the current review followed the requirements for systematic reviews and meta-analysis according to the protocols of the PRISMA-P guidelines [6].

### Inclusion and Exclusion Criteria

The scientific search included combinations of keywords and terms related to coronavirus disease (COVID-19), such as “SARS-CoV-2”, “Viral infection”, “Post-COVID-19”, “Long-COVID-19”, “ARDS”, “Inflammation”, “Cytokine storm”, “Oxidative stress”, “Pathogenesis of CIVID-19”, “Free radical damages”, “Mitochondrial dysfunction”, “mROS”, “COVID-19-induced endothelial damage”, and “COVID-treatment and prevention”.

Our study included 186 publications dated before September 1st, 2023 in the following areas: (1) “Oxidative stress-related diseases” (scientific publications from 1991 to 2023), and (2) “Oxidative stress, endothelial damage and mitochondrial dysfunction in SARS-CoV-2 infection” in the period from 2019 to 2023. We excluded letters, commentaries, preprints, non-English-language articles, and scientific publications presenting other viruses (MERS-CoV, RSV, ebola, HIV, etc.). Clinical and preclinical studies (reviews, articles, case reports, etc.), that showed the role of oxidative stress and redox disbalance in other diseases than COVID-19 were excluded. The process of data analysis and study selection is presented in Figure 1.

## 3. Oxidative Stress, Free Radicals, and Redox State Conception

The concept of “Oxidative stress” was first formulated by Sies in 1985, according to whom “Oxidative damage caused by reactive oxygen species is called “oxidative stress” [7,8]. From then until today, “Oxidative stress” has caused considerable interest in the scientific community and is a prerequisite for fundamental studies in various biomedical fields. Oxidative stress is a process of altered redox homeostasis induced by psychological, physiological, or environmental stressors. Redox OS is an imbalance between antioxidants and pro-oxidants in favor of oxidants. OS is a phenomenon that involves abnormally high levels of reactive oxygen species and reactive nitrogen species (ROS and RNS), impaired antioxidant capacity, and loss of redox control in the body [9,10].

A growing number of studies have shown that OS is a major player in the onset and progression of diseases [11,12,13,14] and various viral infections, such as human respiratory syncytial viruses, rhinoviruses, etc., with COVID-19 being no exception [15,16]. In general, all patients with viral infections are affected by chronic oxidative stress [17]. During the early stages of SARS-CoV-2, ROS production helps to activate immune cells and inhibit viral replication. However, in severe COVID-19 cases, uncontrolled production of ROS and OS can further exacerbate inflammation to cause tissue damage and multiple organ failure [18]. As the infection progresses, the depletion of the antioxidants ascorbate, glutathione, etc., a reduction in the body’s antioxidant capacity, and the deterioration of oxidative status are observed [19]. A postmortem examination of lung tissue from patients with viral infection showed a high level of oxidized DNA, lipids, and proteins [20]. Furthermore, the increased expression of NO synthase-2 (NOS2) and the high level of nitrated proteins are clear evidence of marked oxidative and nitrosative stress [21].

### 3.1. Free Radicals Reactive Oxygen Species and Reactive Nitrogen Species (RONS)

The term “Free Radicals” can be broadly defined as any molecular species that contains an unpaired electron and is capable of independent existence. In the last decade, many researchers have confirmed that RONS are not only mitochondrial and enzymatically generated byproducts, but also important signaling molecules required to regulate the immune response. Within physiological limits, free radicals play a basic role in various biological processes such as cell signaling and regulation of redox homeostasis [22]. Reactive oxygen and nitrogen species (RONS) play a role in the pathophysiology of infectious processes, including viral infections. RONS are produced by immune cells (macrophages and neutrophils) as part of the host’s defense against viral infections. They act as cytotoxic molecules that damage viral protein, nucleic acids, and lipid membranes of the virus. In this way, they inhibit viral replication and spread, and serve as signaling molecules in the activation of the immune response [23]. Despite their important physiological role, RONS are major OS inducers in the body. While RONS can be beneficial in limiting viral replication, an excessive or uncontrolled production of RONS can lead to oxidative stress and damage to host cells. This oxidative stress can contribute to tissue damage and inflammation seen in many viral infections [24]. It is well known that ROS and RNS are byproducts of oxygen metabolism in biological systems, and their excessive production can have damaging effects on various cellular components, including proteins, by modifying specific amino acid residues, fragmentation of peptide chains, changing the electrical charge of amino acid residues, enzyme inactivation, etc. Their excessive production can damage cellular components, cause damage to mitochondria and the endothelium, disrupt the function of lymphocytes and macrophages, and increase inflammation [25].

As a rule, ROS and RNS are defined as unstable oxygen and nitrogen containing redox-active molecules with one or more unpaired electrons. Among the most important representatives of this class of oxidants are the superoxide anion radical (O_2_•^−^), the hydroxide (•OH) and alkoxyl (•OR) radical, peroxide radicals (•OOR), as well as nonradical species—hydrogen peroxide (H_2_O_2_), nitric oxide (NO), and the peroxynitrite anion (ONOO^−^) [24]. RONS react rapidly with various macromolecules in the cell, with the superoxide radical considered to be a significant initiator of oxidative cell damage and a major precursor of secondary oxidants [26]. Environmental pollutants, poor diet, alcohol consumption, smoking, hypoxia, or malnutrition are among the factors leading to the formation of RONS and OS [27]. Other sources of ROS are solar radiation (UV radiation) and the use of drugs or other exogenous substances. Infections, ischemia-reperfusion (I/R) injury, and various inflammatory processes also lead to increased levels of RONS [26].

### 3.2. Redox State Change, Oxidative Stress, and ROS Generation in COVID-19-Related ARDS

The term “redox status” serves as a general characterization of a connected set of oxidation–reduction reactions and a complex description of the “redox environment” in cells and tissues [28]. It describes the redox state of complex biological systems and multiple redox processes, summarizing the reducing potential and capacity of the organism as a whole [29]. A remarkable insight into redox biology is the understanding that redox reactions are used in living cells in the basic processes of redox regulation, collectively called “redox signaling” and “redox control” [7]. The intracellular redox state is a dynamic system that can be changed by various external and internal factors. A healthy organism is in a state of equilibrium characterized by the maintenance of physiological levels of ROS by intracellular reductants [30]. Several studies show the continuation of constantly high levels of ROS and their indirect generation by xenobiotics, pollutants, or others. Intoxicants initiate oxidative damage to various macromolecules and cellular components, thereby contributing to the development and progression of many neurodegenerative, cardiovascular, and endocrine diseases, inflammation, aging, and cancer [31]. For example, in patients with COVID-19, the virus can cause an increase in ROS production and a decrease in antioxidant levels, leading to a redox imbalance and contributing to the development of severe symptoms, such as acute respiratory distress syndrome (ARDS) [32].

### 3.3. Acute Respiratory Distress Syndrome

Acute respiratory distress syndrome (ARDS) is characterized by a massive loss of lung function and is one of the most prominent symptoms of severe coronavirus infection [33]. The progression of ARDS begins with an alveolar-capillary injury, which is characterized by inflammation, apoptosis, necrosis, and increased capillary permeability, and leads to the development of pulmonary edema and proteinosis. Alveolar edema in turn reduces gas exchange, which causes hypoxemia [34]. Severe inflammation and damage to lung tissue also include symptoms such as shortness of breath, rapid breathing, and low blood oxygen levels and can lead to respiratory failure. Lung segments can be severely affected, reducing regional lung compliance and oxygenation [35]. It has been shown that the rapid rise in the levels of the inflammatory cytokines IL-6, IL-8, and TNF-α in SARS infections is associated with the development of acute respiratory distress syndrome [33,36]. In severe COVID-19, ARDS is a combination of viral pneumonia and severe respiratory syndrome and defined as a predictable serious complication that requires early recognition. The entry of the SARS-CoV-2 virus into the respiratory system and the subsequent initiation of a strong inflammatory response lead to the breakdown of the alveolar–capillary barrier [37]. In the acute stage of infection, ARDS causes lung damage that includes the formation of hyaline membranes in the alveoli, followed by interstitial expansion, edema, and fibroblast proliferation with typical pathological changes characterized by diffuse alveolar parenchymal damage [38]. The accumulation of protein-rich fluid in the alveolar and interstitial spaces causes pulmonary surfactant inhibition [35]. Lung morphology in ARDS reflects the rapid evolution from interstitial and alveolar edema to fibrosis, resulting from alveolar-capillary unit damage [39]. Pathological features fully correspond to the effects of multiple damaging stimuli and the complex interaction of inflammatory mediators with alveolar epithelial and capillary endothelial cells. In ARDS, there is a loss of the normal alveolar–capillary barrier. Fluids in the lungs are more difficult to reabsorb, which is compounded by the overproduction of proinflammatory cytokines and by mechanical ventilation, which involves inflating the lung by delivering large volumes of high-pressure oxygen [40]. Patients placed on mechanical ventilation face another paradox. They have shown progression of lung damage due to alveolar overdistension (barotrauma) [41]. A high respiratory rate and inspiratory flow increase total lung stress, which can occur at both high and low lung volumes [42]. X-ray images of patients with COVID-19-induced pneumonia possess unique and distinctive features that are defined by an opaque ground-glass-like appearance [39,43]. Diffuse alveolar damage resulting from the disruption of alveolar cells and endothelium is characterized by the presence of hyaline membranes, interstitial and alveolar edema, and an inflammatory infiltrate [44]. Its physiological manifestation is expressed in severely impaired gas exchange, with refractory hypoxemia and hypercarbia, intrapulmonary shunt, and reduced functional lung surface area [45]. In most cases, the SARS-CoV-2 virus causes a mild infection of the respiratory tract that does not require medical intervention. However, in some patients, the disease is severe and demands hospitalization before developing a heavy clinical picture and complications [46,47,48].

ROS production can cause oxidative damage to lung tissue and trigger an inflammatory response leading to massive lung dysfunction and impaired oxygen exchange. Redox imbalance further contributes to the progression of ARDS and respiratory failure development [49].

## 4. Role of the Immune System, Inflammatory Cytokines, and OS in SARS-CoV-2-Induced Lung Injury

Various pathogens, autoimmune diseases, malignancies, and certain therapeutic interventions can lead to life-threatening systemic inflammatory reactions. They have a common feature expressed in the excessive activation of immune cells and massive release of cytokines. This dysregulated inflammatory response results in a self-reinforcing feedback loop that can be fatal to the host [50]. It is important to note that COVID-19 is a complex disease, and redox imbalance is only one aspect of the development and pathogenesis of the infection. In addition, factors such as viral load, immune response, and individual body response also contribute to the severity of the disease [51,52] (Figure 2).

COVID-19 initiates hyperinflammation known as an excessive immune response called a “cytokine storm”. Cytokine storm is an umbrella term and encompasses various events leading to multiple organ failure and death in patients with COVID-19 [53]. Indeed, a cytokine storm is defined as an acute, local, systemic, and uncontrolled release of proinflammatory markers [54]. The activation of an inflammatory response to viral invasion is accompanied by the overproduction of large amounts of cytokines such as interleukin-6 (IL-6), tumor necrosis factor-alpha (TNF-α), etc. [55]. Secondary hemophagocytic lymphohistiocytosis (sHLH) is a hyperinflammatory syndrome characterized by fulminant and fatal hypercytokinemia and progressive multiple organ failure [56]. Its main features include persistent fever, cytopenia and hyperferritinemia, and lung injury (including ARDS). The pathogenetic mechanisms of sHLH include defective cytotoxicity and uncontrolled T-cell activation, which induces hyperinflammation [57]. The cytokine storm profile in COVID-19 resembles sHLH and is characterized by increased levels of interleukins (IL-1β, IL-6, IL-8, IL-18), interferon-γ (IFNγ), granulocyte colony-stimulating factor (G-CSF), monocyte chemoattractant protein 1 (MCP-1/CCL2), macrophage inflammatory protein (MIP), and TNF-α. Elevated ferritin and IL-6 levels are considered predictors of fatal outcomes, confirming that mortality may be due to virus-induced hyperinflammation [58]. Various studies have suggested that proinflammatory cytokines (mainly IL-6) are responsible for the acute lung injury seen in COVID-19 [33]. According to Pustake and colleagues, strong expression of IL-6 is higher in patients with SARS-CoV-2 compared to those with MERS-CoV infection [59] and is associated with disease severity [60,61]. Interleukin-6 is an important member of the cytokine network and a central mediator in the cytokine release syndrome [62,63]. It mediates multiple signaling pathways and is involved in the regulation of cell proliferation, differentiation, apoptosis, angiogenesis, and metastasis. At the same time, abnormal IL-6 synthesis plays a pathological role in chronic inflammation and autoimmune diseases [64,65]. In infectious conditions or tissue injury, IL-6 is rapidly produced by various cells such as monocytes, macrophages, endothelial cells, fibroblasts, etc., which promotes host defense during the acute phase of infection [33]. In acute inflammation, IL-6 stimulates the liver to synthesize a large number of proteins, including CRP, serum amyloid A (SAA), and others [65].

The management of cytokine storm is oriented toward the use of anti-inflammatory therapies with corticosteroids or monoclonal antibodies that target specific cytokines [66]. Antioxidant and oxygen therapy are used to manage the severe respiratory symptoms that occur as a result of the inflammatory response in the body [67]. All these therapeutic approaches are aimed at reducing the production and release of proinflammatory markers, lowering ROS levels, and increasing oxygenation in the body, thereby mitigating the harmful effects of infection [68]. ROS generation and the activation of innate immunity are important steps in the defense mechanism against various viral infections and respiratory diseases [69]. Indeed, influenza infections are accompanied by the overproduction of ROS and RNS in the alveolar epithelium [17]. An excessive production of O_2_•^−^, H_2_O_2_, and •OH, along with decreased expression of antioxidant enzymes, leads to over-expression of proinflammatory cytokines (IL-6, TNFα, IL-1β, etc.) [70] and activating factor 2 (Nrf2) [71]. In the initial phase of the disease, after the virus enters the respiratory tract, its replication begins, followed by an immune response involving macrophages and dendritic cells [72]. COVID-19 manifests itself with characteristic changes in biochemical indicators, such as high levels of D-dimer, C-reactive protein (CRP), etc., which can be considered predictors of severe disability, multiple organ failure, and fatal outcomes [73,74,75,76]. Studies of patients with coronavirus pneumonia have shown decreased hemoglobin levels [76,77], elevated serum ferritin, increased erythrocyte sedimentation rate, hypoalbuminemia, and acidosis [58]. In reply to the uncontrolled immune response and high levels of ROS, erythrocyte damage occurs. This pathological process is associated with the release of free iron into the bloodstream, which through Fenton reactions in the presence of H_2_O_2_ leads to the formation of the highly toxic •OH. Hydroxyl radicals convert soluble plasma fibrinogen into dense, enzyme-resistant fibrin clots, worsening the patient’s condition. During disease progression, a disseminated intravascular coagulopathy characterized by a marked increase in fibrin/fibrinogen degradation products is observed [46]. On the other hand, H_2_O_2_ indirectly enhances the regulation of inflammatory cytokines by activating the NF-κB pathway and activates macrophages, neutrophils, and endothelial cells through NADPH oxidase, which causes the generation of an additional amount of O_2_•^−^ and H_2_O_2_. Adding to the overall picture, the superoxide radical reacts with iNOS-produced nitric oxide to produce the highly mitochondrially toxic ONOO^−^. The formed ONOO^−^ causes damage to the mitochondria, making them unable to use oxygen, despite normal tissue saturation. In addition, the virus inhibits Nrf2, responsible for the increase of enzymatic antioxidants, and thus reduces the endogenous antioxidant defense system in the body [71,72].

## 5. Role of the Mitochondria in the Pathogenesis of COVID-19

Violations in mitochondrial dynamics, initiation of mitochondrial dysfunction, and redox imbalance in the body contribute to a wide range of pathological conditions, such as inflammation, cardiovascular and neurodegenerative diseases, metabolic disorders, cancer, etc., called “free radical diseases”.

Mitochondria play a central role in the regulation of cellular bioenergetics. They are dynamic organelles responsible for maintaining the energy needs of the cell and undergo frequent changes in their size, shape, and distribution. They participate in the regulation of membrane potential, the synthesis of heme and hemoglobin, cell proliferation, etc. [78]. Mitochondrial redox control is important not only for oxidative phosphorylation, ATP synthesis, calcium homeostasis, thermogenesis, apoptosis, and ROS production, but also for the maintenance of redox balance in cells. Mitochondrial gene mutations that underlie various diseases can disrupt mitochondrial energy metabolism, mitochondrial bioenergetics, and biosynthesis and serve as a trigger for mitochondrial “retrograde signaling” in the nucleus [79].

In healthy humans, mitochondrial free radical generation is tightly regulated enzymatically by mitochondrial superoxide dismutase (mSOD2) [80], GPx [81], CAT [82], as well as by non-enzymatic antioxidants [83]. Mitochondrial antioxidant systems exert strict control over the levels of primary-produced ROS; however, disturbances in redox control identify the mitochondria as the main source of intracellular oxidants [84]. Indeed, the intracellular redox balance is a vulnerable system that is altered by many factors, one of the most important of which involves a change in normal mitochondrial dynamics, in response to changes in mitochondrial shape, size, distribution, and movement within cells. These dynamic fluctuations are essential for maintaining normal mitochondrial function and cellular homeostasis and are regulated by a complex interplay of fission, fusion, and mitophagy [85]. Pathological changes in mitochondrial dynamics can be caused by the overproduction of ROS and the mitochondrial dysfunction they initiate. As a result, oxidative DNA damage processes are promoted and a prerequisite is created for impaired redox regulation, which contributes to a wide range of pathological changes in cells [86]. For example, a major pathophysiological feature of cardiovascular diseases is the overproduction of mROS, which are involved in the initiation of structural and functional changes in cardiomyocytes [87]. A similar picture is observed in aging, diabetes [85], atherosclerosis, neurodegeneration [33], and cancer [88], where an overproduction of ROS in mitochondria and changes in mitochondrial dynamics play a key role [89].

### Mitochondrial Imbalance and Endothelial and Mitochondrial Dysfunction in COVID-19

Mitochondria are the “powerhouses” of cells [90], responsible for producing the energy molecule adenosine triphosphate (ATP). High oxidative stress can damage mitochondrial components, including DNA, proteins, and lipids, which leads to impaired mitochondrial function, reduced ATP production, and bioenergetic collapse [72,91]. In the context of endothelial cells, this can result in impaired cellular functions, including the synthesis of nitric oxide (NO). It is known that NO is essential for maintaining the dilation of blood vessels and regulating blood flow. When endothelial cells experience mitochondrial dysfunction and energy production disruption, they are less able to produce nitric oxide and other vasoactive substances [92]. This leads to endothelial dysfunction, characterized by impaired vasodilation, increased vascular inflammation, and a prothrombotic state [93]. This dysfunction is a critical factor in the development of various acute conditions [35,94,95], etc., and it can further exacerbate OS itself, creating a vicious cycle [96].

The functional changes that occur in the mitochondria are suggested to be an important factor in the progression and development of COVID-19 [97]. Several studies have reported that SARS-CoV-2 can lead to mitochondrial dysfunction in different cells. The virus induces a significant reduction in mitochondrial membrane potential and increased ROS production in lung epithelial cells, which can lead to OS and lung tissue damage [98,99].

The involvement of lung epithelial cells, endothelial cells, and immune cells is associated with a severe COVID-19 clinical picture associated with respiratory failure, myocardial damage, the development of ARDS, and multiple organ failure [100]. SARS-CoV-2 can compromise mitochondrial dynamics by damaging mitochondrial DNA (mtDNA) and altering mitochondrial membrane potential and calcium homeostasis, which disrupts the redox balance in the organism [84]. Direct evidence for this is found in the higher levels of mtDNA in the blood of patients with COVID-19, which is due to increased mitochondrial stress (mtOS) and mitochondrial dysfunction [85].

Mitochondrial reactive oxygen species (mtROS), released during metabolism, are signaling molecules in maintaining cellular homeostasis and regulating inflammatory pathways. Under normal conditions, mitochondria produce small amounts of oxidants (ROS such as superoxide and hydrogen peroxide), which result from the incomplete reduction of oxygen [101]. On the other hand, abnormal levels of mROS that exceed mitochondrial antioxidant capacity lead to Ca^2+^ release from the endoplasmic reticulum (ER), which can disrupt mitochondrial dynamics, cause the consumption of mitochondrial components, mitochondrial rupture, and mtDNA release, and contribute to apoptotic and necrotic cell death. Under such conditions, oxidative and nitrosative stress deplete the cell’s mitochondrial reserve capacity, leading to ongoing or recurrent stress and a vicious inflammatory cycle [102]. Mitochondrial reactive oxygen species are potent oxidants that participate in electron transfer reactions leading to the formation of highly reactive ROS. Binding of SARS-CoV-2 spike proteins to the ACE2 receptor leads to the downregulation of ACE2, which initiates Ang II accumulation and NADPH oxidase activation. In turn, NADPH oxidase leads to the formation of O_2_●^−^, which reacts with NO to form ONOO–. Excess mtROS induces oxidative and nitrosative damage to the ETC and the inactivation of ETC complexes, leading to an increased production of inflammatory cytokines [101,103].

The formation of excessive mROS levels can lead to mitochondrial dysfunction due to impaired efficient electron transfer and direct oxidative damage to telomeres, promoting inflammation [104] and initiating endothelial dysfunction. Thus, SARS-CoV-2-induced OS and disturbances in mitochondrial dynamics promote chronic inflammation and endothelial damage even after viral particles have left the body [105].

## 6. Endothelial Injury and Thrombosis: Disseminated Intravascular Coagulopathy (DIC) and Venous Thromboembolism in COVID-19

Damage to the epithelial lining of the airways through the release of cytokines and chemokines from immune cells is a hallmark of COVID-19. The underlying mechanisms of endothelial damage in COVID-19 are still being studied, but viral replication, the immune response, inflammatory mediators, and mitochondrial dysfunction are believed to be major contributors. Endothelial damage is a potential complication of COVID-19 and leads to impaired blood flow, increased risk of thrombosis, and ARDS. The severity of epithelial damage in COVID-19 can vary widely, with factors such as advanced age, chronic disease, and compromised immune status influencing the degree of epithelial involvement [106].

### 6.1. Oxidative Stress and Endothelial Dysfunction in Liver Sinusoidal Endothelial Cells

In the context of COVID-19, there have been reports and studies indicating that the virus can infect endothelial cells, which can lead to endothelial dysfunction and contribute to the complications seen in severe cases of liver disease [107]. Liver sinusoidal endothelial cells (LSECs) are a specialized type of endothelial cell found in the liver’s sinusoids. They possess unique fenestrations (small pores) in their cytoplasm, which allow for an efficient exchange of substances between the bloodstream and liver parenchymal cells (hepatocytes) and are considered the gatekeepers of liver homeostasis [108]. Many infections or inflammation can induce LSEC dysfunction, which can lead to disruptions in liver function and contribute to damage [109]. As an example, SARS-CoV-2 infection can trigger an inflammatory response and result in endothelial dysfunction, which may contribute to liver-related complications in COVID-19 patients [110]. In the case of COVID-19, there is evidence to suggest that the virus can infect endothelial cells, including LSECs. COVID-19 has been associated with an increased inflammatory response, which can lead to OS, and this can affect LSECs in several ways. High levels of RONS and redox imbalance can lead to damage to the fenestrae, the small pores in LSECs [111]. This damage can alter the structure and function of the fenestrae, reducing the efficiency of nutrient and oxygen exchange between the bloodstream and hepatocytes and causing an ineffective regulation of blood vessel tone modulation in the liver [112]. This can contribute to changes in blood pressure within the liver and impact overall liver function. Also, OS can trigger an inflammatory response within the liver, further exacerbating sinusoidal endothelial dysfunction and contributing to organ damage manifested as elevated liver enzymes (e.g., alanine aminotransferase (ALT), aspartate aminotransferase (AST), gamma-glutamyl transferase (GGT), alkaline phosphatase (ALP)) and/or total bilirubin (TBIL)) [107].

### 6.2. Endothelial Damage of the Vascular Layer as a Result of Oxidative Stress in COVID-19

The endothelium is a layer of cells that lines the inner surface of blood vessels and plays an important role in regulating blood flow and preventing thrombotic events. Simultaneously, it actively participates in the production of various cytokines and adhesion molecules and is involved in key processes such as angiogenesis, coagulation, and regulation of vasomotor tone [113]. Endothelial dysfunction is characterized by increased permeability and edema formation, disruption of the balance between vasodilators and vasoconstrictors, and increased expression of adhesion molecules, as well as the release of ROS, etc. [114].

Chronic inflammation and OS are associated with the pathogenesis of various cardiovascular and cerebrovascular diseases such as atherosclerosis, hypertension, etc.; at the same time, they are considered to be major players in endothelial dysfunction, although the exact mechanisms are still not fully understood [115,116]. Endothelial dysfunction is thought to play a major role in the pathogenesis of COVID-19 and is associated with microangiopathy, pulmonary vascular changes, red blood cell microaggregates and platelet activation, and alveolar capillary microthrombi and endothelial damage [117]. In experimental models of hypertension, higher levels of ROS in blood vessels and oxidative damage to the vascular endothelium have been found [118]. Histopathological findings show that the pathology of COVID-19 includes not only hyperinflammation and cytokine storm, but also pronounced coagulopathy, impaired mitochondrial dynamics, and changes in vascular tone [119].

A spike in ROS levels initiates cellular damage and disrupts mitochondrial dynamics of platelet mitochondria and platelet function, which may induce hypercoagulation and thrombosis in COVID-19 [120]. On the other hand, the accumulation of dysfunctional mitochondria creates a prerequisite for the formation of free mitochondrial reactive oxygen species (mROS) and intermediate ROS products [121], an increase in nicotinamide adenine dinucleotide phosphate oxidase (NADPH oxidase) activity, and the inhibition of antioxidant signaling pathways [89]. Mitochondrial dysfunction is not limited to cellular mitochondria, but can also be observed in those free or membrane-enclosed platelets or vesicles [120]. The direct interaction of SARS-CoV-2 with endothelial cells increases the expression of inflammatory mediators such as IL-6, TNF-α, etc., which together with high levels of oxidants lead to endotheliitis, cell lysis, and apoptosis, disrupting the integrity on the vascular wall [122]. The endothelium performs a fine control of the vascular tone, participating in the maintenance of tissue hemostasis and vascular permeability. Endothelial NOS (eNOS) is mainly expressed in the cardiovascular system and is responsible for the production of nitric oxide (NO), which can interact with ROS and increase the formation of peroxynitrite. A disturbed NO/ROS balance causes increased vasoconstriction, oxidation, inflammation, thrombosis, and proliferation in the vascular wall and is one of the hallmarks of endothelial dysfunction [123].

SARS-CoV-2 viral pneumonia causes an overactivation of the immune response in lung tissue, which is almost always accompanied by OS, loss of redox control, and compromised mitochondrial function and dynamics, with subsequent dramatic endothelial damage [17] (Figure 3).

The SARS-CoV-2 virus can directly infect the endothelial cells of the vasculature, which potentiates cell damage and apoptosis [124]. This leads to a decrease in the antithrombotic activity of the vascular layer and an increase in factor VIII and von Willebrand factor (vWF). Elevated vWF reflects endothelial damage and indicates a high degree of platelet aggregation [125]. The close relationship between vWf and the processes of thrombus formation (thrombogenesis) or atherogenesis also suggests that high levels of vWf may be a useful indirect indicator of thrombosis [126]. Prevention of venous thrombosis is an important component of the complex and comprehensive treatment of COVID-19 and the post-COVID syndrome. Despite thromboprophylaxis in patients with coronavirus infection, the incidence of thromboembolic events is increasing [127].

### 6.3. DIC and Thrombosis in COVID-19 Cases

Thromboembolic events are associated with acute arterial ischemia due to obstruction of arterial vessels and occur despite the use of thromboprophylaxis and anticoagulant therapy [128]. Also, the overactivation of the coagulation system and inflammation in ARDS may increase the risk of pulmonary embolism, as the prevalence of extensive thrombosis and alveolar capillary microthrombi is significantly increased in COVID-19 [129,130]. Patients over 70 years of age are defined as particularly at risk for the development of arterial and venous thromboembolic events [131].

Disseminated intravascular coagulopathy (DIC) is a hemostatic disorder caused by various conditions such as sepsis, pancreatitis, trauma, and pregnancy [132], etc., as well as in patients with COVID-19 [133,134]. DIC occurs due to the abnormal activation of a cascade of reactions involved in blood clotting, leading to the activation of fibrinolytic mechanisms and deposition of fibrin in small vessels. Coagulation proteins and platelets may be depleted during the ongoing prothrombotic and fibrinolytic processes, leading to hemorrhage. Thus, in DIC, spontaneous bleeding and thrombosis can occur simultaneously, as an acute or chronic condition in different diseases [135].

The high mortality of patients with COVID-19 is due to various complications, including venous thromboembolism and disseminated intravascular coagulation (DIC) as a consequence of a systemic inflammatory response in the body [136,137]. Infection can lead to immunologically mediated vasculitis, expressed as an acute inflammatory reaction of the vascular layer, and causes thrombotic changes. Major risk factors include stasis, endothelial damage, and a hypercoagulable state known as Virchow’s Triad [138]. Milovanovic and colleagues consider endothelial damage, microvascular inflammation, and endotheliitis as the basis for severe COVID-19 infection and predictors of high mortality [139].

SARS-CoV-2 infection begins when viral spike proteins bind to ACE2, which initiates viral endocytosis into the endoplasmic reticulum. This increases the amount of Ang II and activates the enzyme NADPH oxidase. As a result, the production of superoxide radicals, hydrogen peroxide, and peroxynitrite is increased. Increased production of ROS and RNS affects the mitochondria, leading to the formation of mROS through oxidative and nitrosative damage to the mitochondrial electron transport chain. Mitochondrial ROS mediate various signaling pathways, increase the expression of inflammatory cytokines (IL-6, IL-1β, TNF-α, etc.), and increase the risk of thrombosis in patients with COVID-19 [140] (Figure 4).

Increased ROS formation leads to a disruption of the balance between antioxidants and oxidants, the activation of NADPH oxidase, and reduced NO bioavailability [118]. Low NO levels predispose the vasculature to a proinflammatory and prothrombotic state that can potentiate endothelial dysfunction [117]. As a result of the expression of proinflammatory cytokines and adhesion molecules, there is increased vascular permeability and hypertrophy of the vascular layer [141], loss of integrity of the epithelial–endothelial barrier, and passage of fluids and proteins to extravascular compartments, shortening of the half-life of proteins such as albumin [142,143]. A significant increase in IL-6 induces tissue factor (Factor III) expression in monocytes and macrophages, leading to thrombin generation [133]. In addition, inflammation contributes to thrombosis through endothelial damage and maintenance of a hypercoagulable state. Thrombi from extracorporeal membrane oxygenation contain more neutrophils than blood clots in other diseases [143]. Thrombosis and DIC are vascular events that are actively involved in the progression of COVID-19 and represent a coagulation/fibrinolytic abnormality manifested by macro- and microthrombotic events. The likelihood of the occurrence of COVID-19 thrombosis increases with increased markers of coagulation and fibrinolysis [144,145].

## 7. Oxidative Stress and Endothelial Dysfunction in Post-Acute Sequelae of COVID-19 or Long-COVID-19 Syndrome

Increasing scientific evidence presents long-COVID-19 as a multifactorial disease involving inflammation, endothelial dysfunction, and a high incidence of thrombotic events, which mainly affects hospitalized COVID-19 patients. Post-acute consequences of COVID-19 (post-COVID-19) syndrome, known as “Long-COVID-19” syndrome, is defined as a multisystem disease, with persistent symptoms that last more than 12 weeks after the diagnosis of acute COVID-19 infection [146]. Approximately 10–30% of non-hospitalized cases [147], 50–70% of hospitalized cases, and 10–12% of vaccinated patients who recovered from mild to moderate COVID-19 were found to continue to suffer from various complications [148]. In general, these include milder symptoms such as fatigue, joint and chest pain, dyspnea, cough, palpitations, alopecia [149], gastrointestinal symptoms, psychological distress, and cognitive dysfunction [150]. Among the more severe manifestations of post-COVID-19 are cardiovascular, thrombotic, and cerebrovascular complications, type 2 diabetes [148], stroke, renal failure, myocarditis, and pulmonary fibrosis [146].

Tsilingiris and colleagues indicated that a chronic low-grade inflammatory response, immune dysregulation, intestinal dysbiosis, persistent endothelial dysfunction and hypercoagulable state, hormonal and metabolic dysregulation, and mitochondrial dysfunction secondary to cellular hypoxia, increased OS, immune dysregulation, or increased inflammation are the main pathogenetic mechanisms for the occurrence of long-COVID-19 syndrome [151]. For example, Stufano et al. investigated the role of oxidative stress in non-hospitalized patients four months after COVID-19. They found that serum MDA levels were significantly higher in patients with COVID-19 compared to healthy controls, suggesting a systemic redox imbalance after the active phase of infection, including in milder cases of SARS-CoV-2 [152]. In the present point, we will briefly highlight the role of OS and specifically mROS in the pathogenesis of long-COVID-19 syndrome and endothelial damage. Endothelial involvement has been suggested to be among the main mechanisms in long-COVID-19 syndrome, especially in severe cases of infection, even when a considerable time has passed since acute illness. The presence of proinflammatory mediators leads to a procoagulant state, an overactivation of coagulation and thrombus formation, increased fibrinogen, vWf, thrombomodulin, and protein S, etc. [153]. Increased levels of vWf, Ang-II, and P-selectin and a decreased activity of Willebrand factor-cleaving protease (VWFCP, ADAMTS-13) further highlight endothelial activation in critically ill patients with COVID-19 [154]. Platelet adhesion, thrombosis, breakdown of the protective glycocalyx, and an impaired vascular barrier are among the most important features of infection [114].

ROS overproduction in viral infections, including COVID-19, stimulates various pathophysiological responses, such as endothelial dysfunction, hypercoagulation, and thrombosis, that contribute to the severity of infection [155]. The binding of the ACE receptor to the SARS-CoV-2 virus leads to high levels of superoxide radicals. In parallel, the activation of macrophages and neutrophils, with the participation of the NADPH oxidase complex, leads to an additional production of superoxide radicals and hydrogen peroxide. In hypoxic COVID-19 patients, ARDS results in decreased oxygen transport to tissues, which increases ROS production and initiates the subsequent expression of inflammatory cytokines, chemokines, and interferons. Concomitantly, inflammatory cytokines can increase OS markers by activating macrophages and neutrophils, creating a vicious cycle of oxidative damage and inflammation [155].

This interplay between oxidative stress and inflammation can induce cumulative oxidative damage and changes in various macromolecules and cellular components [152]. On the one hand, a spike in ROS levels initiates cellular damage and disrupts mitochondrial dynamics [120], and on the other hand, the accumulation of dysfunctional mitochondria creates a prerequisite for the formation of free mROS and intermediate ROS products [89,97,156]. In response, there is a disruption of the intracellular redox balance and activation of redox-sensitive effector pathways, followed by a strong immune response, apoptosis, and necrosis [117,130,157]. Alterations in normal mitochondrial dynamics compromise epithelial barrier function and increase its permeability and susceptibility to infection. This further exacerbates inflammation, creating a vicious cycle between mitochondrial dysfunction and epithelial–endothelial damage, potentiating the development of long-term complications known as post-COVID-19 or long-COVID-19 syndrome. These effects are thought to be due to persistent epithelial damage and inflammation, which may lead to post-mitochondrial dysfunction [158].

## 8. COVID-19: Treatment and Prevention

SARS-CoV-2 is a member of the coronavirus family, which includes viruses that cause mild illnesses like the common cold and more severe diseases like severe acute respiratory syndrome (SARS) and Middle East respiratory syndrome (MERS). SARS-CoV-2 has a characteristic spherical shape with spike-like projections on its surface, which give it a crown-like appearance, hence the name “coronavirus” [159]. CoV-2 viruses have a single-stranded RNA genome, which encodes various proteins. This also includes the spike protein (S protein), which plays a crucial role in its ability to infect human cells. The spike protein interacts with the human ACE2 receptor on cells in the respiratory system, allowing the virus to enter and infect those cells [160,161]. Viral infectivity depends on interactions between components of the host cell plasma membrane and the virus envelope. As SARS-CoV-2 spreads and replicates, it undergoes mutations. Some of these mutations result in the emergence of new variants. These variants can differ in terms of transmissibility, severity of illness, and resistance to immunity from previous infection or vaccination [162].

It is important to note that information about SARS-CoV-2 and COVID-19 is continuously evolving as new research and therapies become available, including the understanding of and treatments for the long-term effects of COVID-19. This point considers, in short, the potential natural antioxidant treatment that could help stem the advance of the SARS-CoV-2 epidemic.

The emergence of COVID-19 has prompted the scientific community to work hard to design vaccines and monoclonal antibodies against SARS-CoV-2 [163]. In addition, severe infection in some patients and the multiple complications known as post-COVID-19 or long-COVID-19 syndrome directed our eyes to the development of new and complex therapies [164]. One of the potential options for limiting cell invasion and replication of SARS-CoV-2 is the application of natural antioxidants [163,165] or combining them with anti-inflammatory therapies [166]. Angiotensin-converting enzyme 2 (ACE2) is a transmembrane glycoprotein that is an important part of the renin–angiotensin system (RAS). It is expressed in alveolar cells, epithelial cells, adipose tissue, the central nervous system, etc., and to a very minor extent in macrophages, monocytes, and T cells. ACE2 acts as a host cell surface receptor, which highlights the role of ACE2 as a “gateway” for multiple viruses including SARS-CoV and SARS-CoV-2 [35,167]. ACE2 mediates virus entry into cells through an interaction between the transmembrane spike (S) glycoprotein of the virus and the N-terminal segment of ACE2 in the target cell [168]. Like the SARS-CoV-2 virus, HHVs attach to the host cell, specifically to the binding receptors of the host cells via several viral glycoproteins [163,169]. Human herpes viruses (HHV) belong to the Herpesviridae family, which includes three subfamilies: herpes simplex virus (type 1 (HSV-1) and type 2 (HSV-2)), varicella-zoster virus (VZV), Epstein–Barr virus (EBV), and human cytomegalovirus (CMV) [170]. They are characterized as infectious double-stranded DNA viruses that periodically reactivate in the human body and can cause a wide range of clinical symptoms and diseases, including cancer. Typically for herpes viruses, they are usually asymptomatic or mildly symptomatic and remain latent for a long period of time in their host [171]. Their reactivation usually occurs due to immune dysregulation and the activation of the interferon cascade, induction of the NLRP3 inflammasome, increased levels of ROS, RNS, and oxidative damage, and decreased antioxidant defense of the body [172]. More and more scientific reports point to the immune response to SARS-CoV-2 as a possible cause for the reactivation of HHVs [163,173,174,175]. Most people infected with SARS-CoV-2 have a mild to moderate form of the infection, which is due to different phenotype and the body’s immune response to the virus [176]. For example, impaired type 1 interferon activity is associated with persistent viral load, leading to an exacerbated inflammatory response and determining the severity of infection. Studies have shown that during COVID-19, patients with latent HHVs experience a reactivation of HHV-6/7, EBV and CMV, and that the coreactivation of these herpesviruses is associated with prolonged duration of mechanical ventilation, extracorporeal membrane oxygenation, and increased ICU stay. Reactivation of HHVs (such as HHV-6, EBV, and CMV) is common in critically ill patients [174]. This reactivation likely facilitates entry of SARS-CoV-2 into epithelial cells, increasing viral load and symptom severity in patients without evidence of prior immune suppression [172]. Thus, the body’s inflammatory response to COVID-19 infection may lead to a vicious circle of coactivation of latent viruses and worsening of the course of SARS-CoV-2 [174].

SARS-CoV-2, the virus responsible for COVID-19, has sparked significant interest in developing antiviral drugs, including those of natural origin [177,178]. While SARS-CoV-2 and HHVs are different types of viruses, the development of antiviral drugs with properties similar to those used against HHVs could offer insights into potential treatments for COVID-19 [171]. Researchers aim to identify compounds that interfere with key stages of viral replication, such as attachment, entry, replication, or assembly [179,180,181,182]. Natural compounds have been a source of antiviral drugs for many years [183]. Research into these natural products may yield potential treatments for SARS-CoV-2, especially drugs that may target common viral replication mechanisms or host cell processes necessary for viral growth. In this aspect, an important approach to preventing SARS-CoV-2 infection may involve new antiviral drugs, including those of natural origin such as cepharanthine, carolacton, etc., [184] with properties similar to those applied for the prevention and treatment of HHVs [163]. Homoharringtonine (Omacetaxine mepesuccinate) and emetine are two natural compounds from the group of alkaloids. They have garnered interest for their potential antiviral properties, including against herpesviruses and emerging viruses like SARS-CoV-2. Homoharringtonine is derived from the bark of the evergreen tree of the Cephalotaxus species and has been investigated for its antiviral properties against various viruses, including herpesviruses and SARS-CoV-2. It is thought to inhibit viral replication by interfering with viral protein translation. Emetine is a naturally derived alkaloid, which has demonstrated antiviral activity against a range of viruses, including some herpesviruses and RNA viruses [163,185]. The parallel combining of multiple antiviral drugs with different mechanisms of action can enhance treatment effectiveness and reduce the risk of resistance development. In this, collaboration between scientists, clinicians, and pharmaceutical companies is vital to advancing our understanding of the virus and developing effective treatments.

### Endothelial Oxidative Stress Treatment and Prevention in the Context of COVID-19

Research on preventing endothelial oxidative stress in the context of COVID-19 is ongoing, and several potential future perspectives and directions can be considered, which can include: (1) new antioxidant therapies; (2) mitochondrial protection; (3) targeted therapy directed towards inflammatory pathways; (4) combination therapies and multidisciplinary collaboration; (5) oxidative stress biomarkers monitoring of COVID-19 and post-COVID-19 patients.

(1) The investigation of antioxidants as therapeutic agents could focus on determining the optimal dosages, timing, and administration routes for these antioxidants in COVID-19 patients to mitigate endothelial dysfunction.

(2) Mitochondrial protection strategies can include maintaining or enhancing mitochondrial health in endothelial cells with drugs or interventions that protect the mitochondria from oxidative damage.

(3) Targeted therapy directed toward inflammatory pathways could investigate drugs or interventions given the close connection between oxidative stress and inflammation. The combination of anti-inflammatory agents with antioxidants can be used to comprehensively address endothelial dysfunction and OS.

(4) Combination therapies and multidisciplinary collaboration directed at investigating the potential synergistic effects of different therapeutic approaches including a combination of antioxidants and antiviral or anti-inflammatory drugs can provide more comprehensive protection against endothelial oxidative stress in COVID-19. Interdisciplinary research in this area is crucial not only for mitigating the acute complications of COVID-19 but also for addressing potential long-term endothelial damage associated with the coronavirus disease.

(5) Oxidative stress biomarker monitoring of COVID-19 and post-COVID-19 patients can involve identifying reliable biomarkers that can assess endothelial oxidative stress in COVID-19 patients. Biomarkers could help in the early detection and monitoring of endothelial dysfunction or the long-term effects of COVID-19 on endothelial function and OS-related damages, which can guide therapeutic decisions. On the other hand, genetic and metabolic profiling could help identify individuals who are more susceptible to critical levels of ROS and RNS and tailor interventions accordingly.

## 9. Conclusions

Despite the variety of measures such as social distancing, various preventive strategies including therapeutic approaches, and the creation of diverse vaccines, the outbreak of COVID-19 and post-COVID-19 complications still hide many mysteries for the scientific community. The discovery of free radicals in biological systems gives a “start” in the study of various pathological processes related to the development and progression of pathologies. From this moment, the enrichment of knowledge about the participation of free radicals and free radical processes in the pathogenesis of cardiovascular, neurodegenerative, and endocrine diseases, inflammatory conditions, and infections including COVID-19 is increasing exponentially.

Oxidative stress is considered to be a major source of ROS-mediated damage in COVID-19 and is an important marker in the severe picture of the disease. The overproduction of free radicals, together with an increased immune response and reduced endogenous enzymatic and non-enzymatic antioxidant defenses lead to a vicious cycle of mutually induced hyperinflammation and massive oxidative damage. Excessive inflammatory responses and abnormal ROS levels during the acute phase of COVID-19 may disrupt mitochondrial dynamics, increasing the risk of endothelial cell damage. The reported abnormal levels of ROS and OS suggest a high degree of structural changes in the macromolecules, the formation of non-functional derivatives, and a high degree of cell component damage in moderate and critical COVID-19. According to the existing scientific data, it can be assumed that the high mortality of SARS-CoV-2 is not only due to infection-related inflammation but also to massive oxidative damage during infection.

In summary, OS and hyperinflammation are critical factors in the pathogenesis of COVID-19 and long-COVID-19 syndrome, and their management is crucial for the treatment and prevention of the disease. Despite the wide range of clinical and biochemical data, there is a need for further expansion of the available information and the standardization of classification criteria and frequency of the clinical spectrum of complications, pathogenetic mechanisms, and prognosis of long-COVID-19, which will allow targeting more effective condition management.

## Figures and Tables

**Figure 1 ijms-24-14876-f001:**
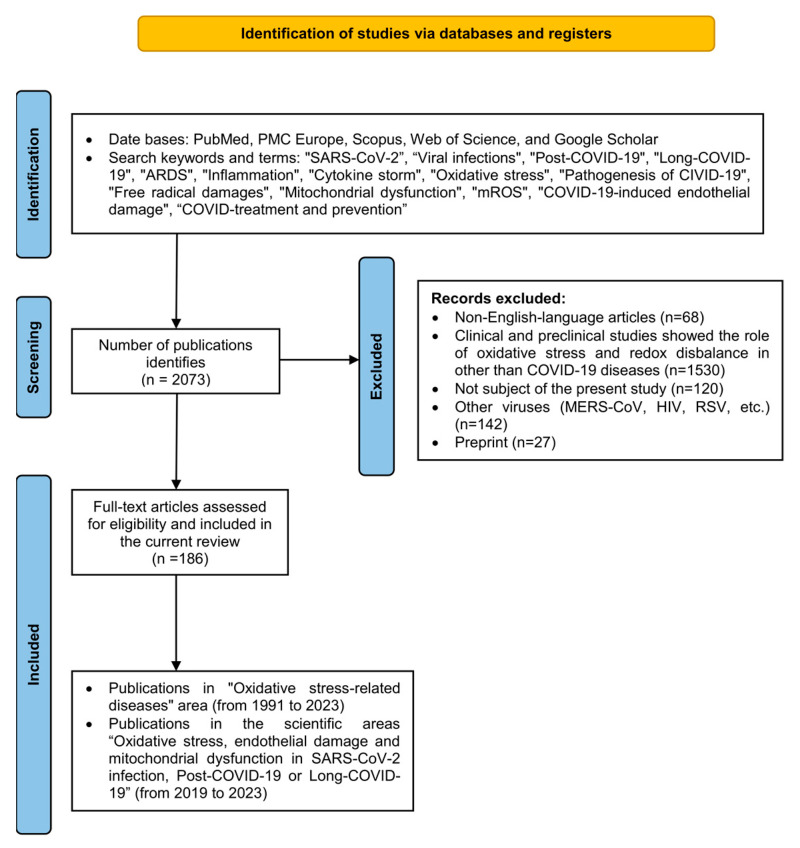
The stages of searching: databases, keywords, applicable criteria, and full-text articles assessed for eligibility and included in the current review.

**Figure 2 ijms-24-14876-f002:**
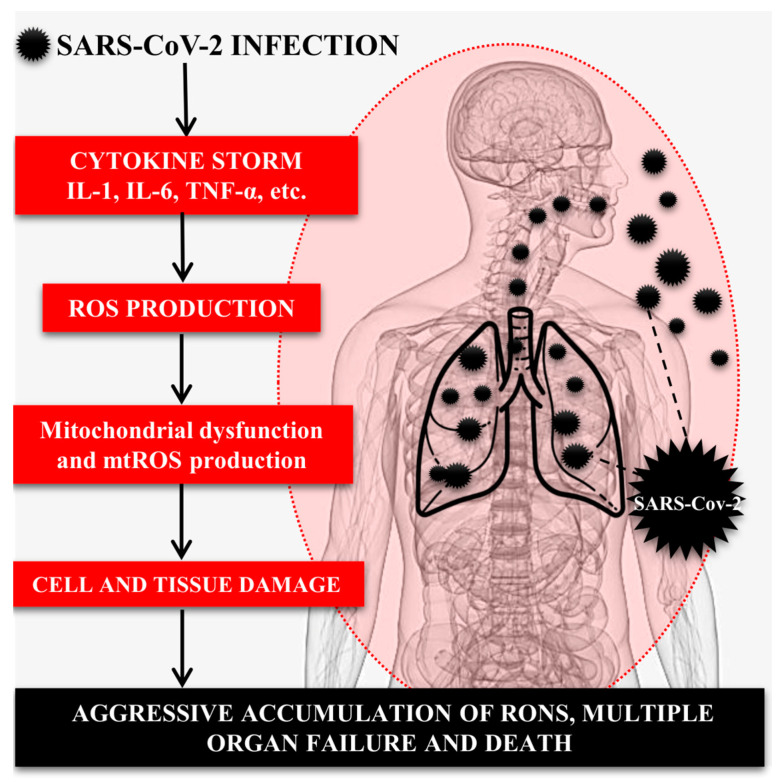
Multiple organ failure in COVID-19 as a function of hyperinflammation and ROS overproduction.

**Figure 3 ijms-24-14876-f003:**
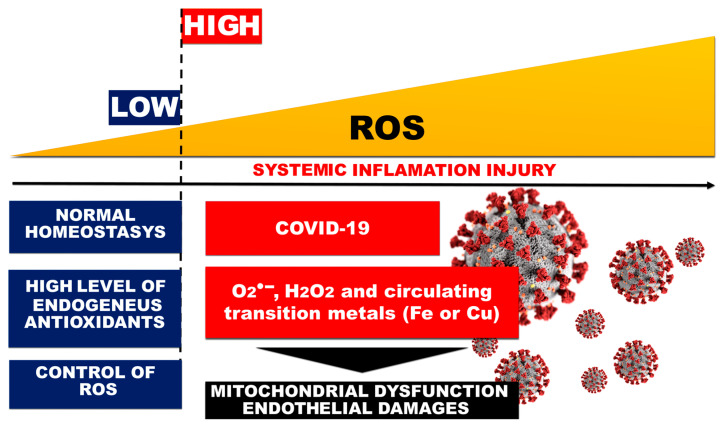
Role of OS and systemic inflammation in COVID-19-related endothelial injury.

**Figure 4 ijms-24-14876-f004:**
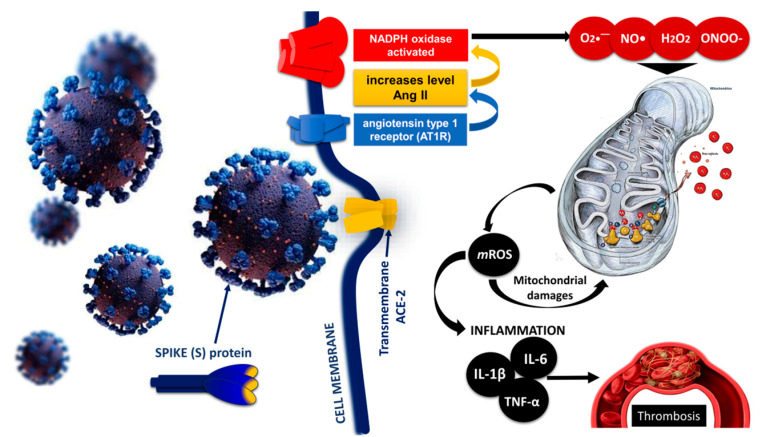
Role of mitochondrial dysfunction, mROS production and inflammation in SARS-CoV-2 induced thrombosis [140] modified by Georgieva E et al. Hyperinflammation increases cytokine production, especially of IL-1, IL-6, and TNF-α, while ROS generated during cytokine storm can damage mitochondria and impair their function through oxidative damage to mtDNA.

## Data Availability

The data presented in this study are available on request from the corresponding author.

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
