# Peer review of "COVID-19 Complications: Oxidative Stress, Inflammation, and Mitochondrial and Endothelial Dysfunction"

_ijms, 2023, doi:10.3390/ijms241914876_

Round 1
Reviewer 1 Report
Although the manuscript delivers a good topic, there are important points that need to be addressed before further consideration.
- I recommend the authors highlight information about the databases used for collecting/extracting the data (for example, Web of Science, Scopus, Google Scholar,..) and what keywords were used during the literature search along with the period of studies included in the review. This ensures that the paper covers all recent and relevant studies. All these points could be highlighted, at least, in the introduction section.
- I recommend the authors discuss the role of inflammation cause by other viral infections such as herpesviruses on COVID-19 patients and how both viruses (herpesviruses and SARS-CoV-2) can induce inflammatory complications. I recommend the authors use the reference (doi: 10.3390/v12040476) to extract the required information.
- I also recommend the authors discuss the current treatment options for preventing or treating the COVID-19 disease.
- The review also lacks an overview of SARS-CoV-2. Please take into consideration this point.
- I recommend the authors double-check the full text for grammatical and typing errors.
I recommend the authors refine the English usage and double-check the full text for grammatical and typing errors.
Author Response
RESPONSES TO THE Reviewers' COMMENTS
We appreciate the reviewers’ comments. All corrections in the manuscript are in red.
Reviewer 1
Point 1. I recommend the authors highlight information about the databases used for collecting/extracting the data (for example, Web of Science, Scopus, Google Scholar,..) and what keywords were used during the literature search along with the period of studies included in the review. This ensures that the paper covers all recent and relevant studies. All these points could be highlighted, at least, in the introduction section.
Answer 1: Done
- Literature Search, Methodology, Inclusion and Exclusion Criteria
For all original articles reporting the association between the redox imbalance, ROS, and RNS, endothelial and mitochondrial dysfunction in COVID-19 and Post-COVID-19 cases, we searched PubMed, PMC Europe, Scopus, Web of Science, Medline, and Google Scholar. The data analysis and preparation of the current review are to the requirements for systematic reviewers and meta-analysis according to protocols PRSMA-P guidelines [6-Moher D et al., 2009]. The scientific search included combinations of keywords and terms related to coronavirus disease (COVID-19), such as "SARS-CoV-2 infection", "Post-COVID-19", "Long-COVID-19", "ARDS", "Inflammation", "Cytokine storm", "Oxidative stress", "Pathogenesis of CIVID-19", "Free radical damages", "Mitochondrial dysfunction", "mROS", "COVID-19-induced endothelial damage", “COVID-treatment and prevention”.
Our study included over 1860 clinical and laboratory studies, articles, literature reviews, and case reports from 1991 to August 2023, whether positive or negative results were reported. Preprint and non-English-language articles were not included in our search. Clinical and preclinical studies (reviews, articles, and case reports) showed the role of oxidative stress and redox disbalance in the initiation and progression of carcinogenesis were excluded (Figure 1).
Point 2. I recommend the authors discuss the role of inflammation caused by other viral infections such as herpesviruses on COVID-19 patients and how both viruses (herpesviruses and SARS-CoV-2) can induce inflammatory complications. I recommend the authors use the reference (doi: 10.3390/v12040476) to extract the required information.
Answer 2: Done
Point 3. I also recommend the authors discuss the current treatment options for preventing or treating the COVID-19 disease.
Answer 3: Done
Point 4. The review also lacks an overview of SARS-CoV-2. Please take into consideration this point.
Answer 4: Done
Point 5. I recommend the authors double-check the full text for grammatical and typing errors.
Answer 5: Done

Reviewer 2 Report
Dear authors, the paper is well written and organized. It lacks of a clear search strategy, and I ask you to clarify your methods. It will be useful for the readers of IJMS not to think of cherry picking misuse.
Author Response
RESPONSES TO THE Reviewers' COMMENTS
We appreciate the reviewers’ comments.
Reviewer
Point 1: Dear authors, the paper is well written and organized. It lacks of a clear search strategy, and I ask you to clarify your methods. It will be useful for the readers of IJMS not to think of cherry picking misuse.
Answer 1: Done
- Literature Search, Methodology, Inclusion and Exclusion criteria
Database and search strategy The number of original articles identified, that reported the association between the redox imbalance, ROS and RNS, endothelial and mitochondrial dysfunction in different acute conditions and chronic diseases was 2073. The databases PubMed, PMC Europe, Scopus, Web of Science, and Google Scholar were used. The data analysis and preparation in the current review followed the requirements for systematic reviews and meta-analysis according to PRISMA-P guidelines protocols [6]. Inclusion and exclusion criteria The scientific search included combinations of keywords and terms related to coronavirus disease (COVID-19), such as "SARS-CoV-2”, “Viral infection", "Post-COVID-19", "Long-COVID-19", "ARDS", "Inflammation", "Cytokine storm", "Oxidative stress", "Pathogenesis of CIVID-19", "Free radical damages", "Mitochondrial dysfunction", "mROS", "COVID-19-induced endothelial damage", “COVID-treatment and prevention”. Our study included 186 publications before September 1st, 2023 in areas, as follows: 1/ "Oxidative stress-related diseases" (scientific publications from 1991 to 2023), and 2/ “Oxidative stress, endothelial damage and mitochondrial dysfunction in SARS-CoV-2 infection” in the period 2019 to 2023. We exclude letters, commentaries, preprints, non-English-language articles, and scientific publications presenting other viruses (MERS-CoV, RSV, Ebola, HIV, etc.) that were not included. Clinical and preclinical studies (reviews, articles, case reports, etc.), that showed the role of oxidative stress and redox disbalance in other diseases than COVID-19 were excluded. The process of data analysis and study selection is presented in Figure 1.

Reviewer 3 Report
The manuscript “COVID-19 complication: Compilation of Oxidative stress, Inflammation, Mitochondrial and Endothelial dysfunction” is a review regarding the role of oxidative stress and free radicals in COVID-19-induced mitochondrial and endothelial dysfunction.
The manuscript might be of interest for the readers. However, authors cover only partially the topic and several critical flaws are present, which prevent the manuscript from being accepted for publication in this form. Authors are strongly encouraged to improve the manuscript accordingly.
1. The manuscript requires an extensive language revision. There are several mistakes and incorrect or confused sentences.
2. The title of the manuscript must be changed, since the word “compilation” has no sense in this context.
3. The authors use the abbreviation “OS” without specifying its meaning at the first use.
4. Line 88: authors use the abbreviation ROS for free oxygen species, which is awkward. Indeed, the universal meaning of the word “ROS” is reactive oxygen species. The same for RNS.
5. Line 110: “RONS are actively involved in the pathophysiology of the infectious process, participating in a powerful mechanism to deal with a wide range of viral infections” this sentence is unclear and should be rewritten.
6. Line 133: again authors use the word “ARDS” without specifying it, since they do it only in the next paragraph.
7. Line 478: I do not understand why there is something written in Cyrillic script.
8. Some title of subparagraphs are sometimes in italics sometimes not. Please reconcile.
9. The figure 1 is somehow confused and does not help the reader. Please rearrange it.
10. Line 278: authors correctly state that Nrf2 counteracts oxidative stress. However they need to specify that Nrf2 is particularly important in endothelium where oxidative stress may lead to disastrous consequences (PMID: 33123312).
11. In the subparagraph 5.1, it should be specified that oxidative stress can induce endothelial dysfunction by affecting the mitochondrial function, since the consequent collapse of the bioenergetic balance of the cell is one of the main effects of oxidative stress (PMID: 34153425).
12. The figure 2 is unacceptable for a scientific manuscript. Contains gross mistakes “HIGT” instead of “HIGH”, “endothelial” instead “endothelial”, “homeostasys", “HIGHT”. It seems it was prepared carelessly. Systematic inflammation injury or systemic?!
13. In the paragraph 6, it must be underlined that the impact of COVID-19 on endothelium can involve also liver vessels. Liver sinusoidal endothelial cells (LSECs) play a crucial role since they normally are gatekeepers of liver homeostasis. Thus, oxidative stress can impair the fenestrae structure and function, inducing endothelial dysfunction in liver (PMID: 31569283).
14. The authors should better explain the future perspectives and directions of the research for preventing endothelial oxidative stress COVID-19-related.
Extensive editing of English language required.
Author Response
Dear reviewer,
Thank You for the very helpful notes. We hope that we have answered properly to all notes correctly. We are applying the list of corrections, and in the manuscript, the corrections are highlighted in yellow:
Point 1. The manuscript requires an extensive language revision. There are several mistakes and incorrect or confused sentences.
Answer 1. Done
Point 2. The title of the manuscript must be changed, since the word “compilation” has no sense in this context.
Answer 2. Done
Point 3. The authors use the abbreviation “OS” without specifying its meaning at the first use.
Answer 3. Done
Point 4. Line 88: authors use the abbreviation ROS for free oxygen species, which is awkward. Indeed, the universal meaning of the word “ROS” is reactive oxygen species. The same for RNS.
Answer 4. Done
Point 5. Line 110: “RONS are actively involved in the pathophysiology of the infectious process, participating in a powerful mechanism to deal with a wide range of viral infections” this sentence is unclear and should be rewritten.
Answer 5. Done
Point 6. Line 133: again authors use the word “ARDS” without specifying it, since they do it only in the next paragraph.
Answer 6. Done
Point 7. Line 478: I do not understand why there is something written in Cyrillic script.
Answer 7. Done
Point 8. Some title of subparagraphs are sometimes in italics sometimes not. Please reconcile.
Answer 8. Done
Point 9. The figure 1 is somehow confused and does not help the reader. Please rearrange it.
Answer 9. Done
Point 10. Line 278: authors correctly state that Nrf2 counteracts oxidative stress. However they need to specify that Nrf2 is particularly important in endothelium where oxidative stress may lead to disastrous consequences (PMID: 33123312).
Answer 10. Done
Point 11. In the subparagraph 5.1, it should be specified that oxidative stress can induce endothelial dysfunction by affecting the mitochondrial function, since the consequent collapse of the bioenergetic balance of the cell is one of the main effects of oxidative stress (PMID: 34153425).
Answer 11 Mitochondria are the "powerhouses" of cells [Yin, F.; Cadenas, E, 2015], responsible for producing the energy molecule adenosine triphosphate (ATP). The high oxidative stress can damage mitochondrial components, including DNA, proteins, and lipids, which leads to impaired mitochondrial function, reduced ATP production, and bioenergetic collapse [Higashi, Y, 2022, Szczesny-Malysiak E et al., 2020 (71)]. In the context of endothelial cells, this can result in impaired cellular functions, including the synthesis of nitric oxide (NO). It is known that NO is essential for maintaining the dilation of blood vessels and regulating blood flow. When the endothelial cells experience mitochondrial dysfunction and energy production disruption, they are less able to produce nitric oxide and other vasoactive substances [Meza, C.A et al., 2019]. This leads to endothelial dysfunction, characterized by impaired vasodilation, increased vascular inflammation, and a pro-thrombotic state [Campagna, R et al., 2021]. This dysfunction is a critical factor in the development of various acute [De Michele M et al., 2023, Szczesny-Malysiak E et al., 2020 (71); Clemente-Suárez, V.J. et sl., 2023] etc., and it self can further exacerbate OS, creating a vicious cycle [Sun HJ et al., 2020].
New references
Yin, F.; Cadenas, E. Mitochondria: the cellular hub of the dynamic coordinated network. Antioxid Redox Signal, 2015, 22, 961-964. https://doi.org/10.1089/ars.2015.6313
Higashi, Y. Roles of Oxidative Stress and Inflammation in Vascular Endothelial Dysfunction-Related Disease. Antioxidants 2022, 11, 1958. https://doi.org/10.3390/antiox11101958.
- Szczesny-Malysiak, E.; Stojak, M.; Campagna, R.; Grosicki, M.; Jamrozik, M.; Kaczara, P.; Chlopicki, S. Bardoxolone Methyl Displays Detrimental Effects on Endothelial Bioenergetics, Suppresses Endothelial ET-1 Release, and Increases Endothelial Permeability in Human Microvascular Endothelium. Oxid Med Cell Longev 2020, 2020, 4678252. https://doi.org/10.1155/2020/4678252.
Meza, C.A.; La Favor, J.D.; Kim, D.-H.; Hickner, R.C. Endothelial Dysfunction: Is There a Hyperglycemia-Induced Imbalance of NOX and NOS? Int J Mol Sci 2019, 20, 3775. https://doi.org/10.3390/ijms20153775.
Campagna, R.; Mateuszuk, Ł.; Wojnar-Lason, K.; Kaczara, P.; Tworzydło, A.; Kij, A.; Bujok, R.; Mlynarski, J.; Wang, Y.; Sartini, D.; Emanuelli, M.; Chlopicki, S. Nicotinamide N-methyltransferase in endothelium protects against oxidant stress-induced endothelial injury. Biochim Biophys Acta Mol Cell Res 2021, 1868, 119082. https://doi.org/10.1016/j.bbamcr.2021.119082.
Clemente-Suárez, V.J.; Redondo-Flórez, L.; Beltrán-Velasco, A.I.; Ramos-Campo, D.J.; Belinchón-deMiguel, P.; Martinez-Guardado, I.; Dalamitros, A.A.; Yáñez-Sepúlveda, R.; Martín-Rodríguez, A.; Tornero-Aguilera, J.F. Mitochondria and Brain Disease: A Comprehensive Review of Pathological Mechanisms and Therapeutic Opportunities. Biomedicines 2023, 11, 2488. https://doi.org/10.3390/biomedicines11092488.
De Michele, M.; Lorenzano, S.; Piscopo, P.; Rivabene, R.; Crestini, A.; Chistolini, A.; Stefanini, L.; Pulcinelli, F.M.; Berto, I.; Campagna, R.; Amisano, P.; Iacobucci, M.; Cirelli, C.; Falcou, A.; Nicolini, E.; Schiavo, O.G.; Toni, D. SARS-CoV-2 infection predicts larger infarct volume in patients with acute ischemic stroke. Front Cardiovasc Med 2023, 9, 1097229. https://doi.org/10.3389/fcvm.2022.1097229.
Sun, H.J.; Wu, Z.Y.; Nie, X.W.; Bian, J.S. Role of Endothelial Dysfunction in Cardiovascular Diseases: The Link Between Inflammation and Hydrogen Sulfide. Front Pharmacol 2020, 10, 1568. https://doi.org/10.3389/fphar.2019.01568.
Point 12: The figure 2 is unacceptable for a scientific manuscript. Contains gross mistakes “HIGT” instead of “HIGH”, “endothelial” instead “endothelial”, “homeostasys", “HIGHT”. It seems it was prepared carelessly. Systematic inflammation injury or systemic?!
Answer 12: Done
Point 13. In paragraph 6, it must be underlined that the impact of COVID-19 on endothelium can involve also liver vessels. Liver sinusoidal endothelial cells (LSECs) play a crucial role since they normally are gatekeepers of liver homeostasis. Thus, oxidative stress can impair the fenestrae structure and function, inducing endothelial dysfunction in liver (PMID: 31569283).
Answer 13:
In the context of COVID-19, there have been reports and studies indicating that the virus can infect endothelial cells, which can lead to endothelial dysfunction and contribute to the complications seen in severe cases of liver disease [Li D et al., 2021]. Liver sinusoidal endothelial cells (LSECs) are a specialized type of endothelial cell found in the liver's sinusoids. They possess unique fenestrations (small pores) in their cytoplasm, which allow for efficient exchange of substances between the bloodstream and liver parenchymal cells (hepatocytes), and they are considered gatekeepers of liver homeostasis [Zapotoczny B et al., 2019]. Many infections or inflammations can induce LSECs dysfunctional, which can lead to disruptions in liver function and contribute to damage [Lafoz E et al., 2020]. An example, SARS-CoV-2 infection can trigger an inflammatory response and result in endothelial dysfunction, which may contribute to liver-related complications in COVID-19 patients [Romano, C et al., 2023]. In the case of COVID-19, there is evidence to suggest that the virus can infect endothelial cells, including LSECs. COVID-19 has been associated with an increased inflammatory response, which can lead to OS and this can affect LSECs in several ways. High levels of RONS and redox imbalance can lead to damage to the fenestrae, the small pores in LSECs [Akkiz, H. et al., 2023]. This damage can alter the structure and function of the fenestrae, reducing the efficiency of nutrient and oxygen exchange between the bloodstream and hepatocytes, and ineffective regulation of blood vessel tone modulation in the liver [Allameh, A. et al., 2023]. This can contribute to changes in blood pressure within the liver and impact overall liver function. Also, OS can trigger an inflammatory response within the liver, further exacerbating sinusoidal endothelial dysfunction and contributing to organ damage manifest as elevated liver enzymes (e.g., alanine aminotransferase (ALT), aspartate aminotransferase (AST), gamma-glutamyl transferase (GGT), alkaline phosphatase (ALP)) and/or total bilirubin (TBIL) [Li D et al., 2021].
Li, D.; Ding, X.; Xie, M.; Tian, D.; Xia, L. COVID-19-associated liver injury: from bedside to bench. J Gastroenterol 2021, 56, 218-230. https://doi.org/10.1007/s00535-021-01760-9.
Zapotoczny, B.; Braet, F.; Kus, E.; Ginda-Mäkelä, K.; Klejevskaja, B.; Campagna, R.; Chlopicki, S.; Szymonski, M. Actin-spectrin scaffold supports open fenestrae in liver sinusoidal endothelial cells. Traffic 2019, 20, 932-942. https://doi.org/10.1111/tra.12700.
Lafoz, E.; Ruart, M.; Anton, A.; Oncins, A.; Hernández-Gea, V. The Endothelium as a Driver of Liver Fibrosis and Regeneration. Cells 2020, 9, 929. https://doi.org/10.3390/cells9040929.
Romano, C.; Cozzolino, D.; Nevola, R.; Abitabile, M.; Carusone, C.; Cinone, F.; Cuomo, G.; Nappo, F.; Sellitto, A.; Umano, G.R.; et al. Liver Involvement during SARS-CoV-2 Infection Is Associated with a Worse Respiratory Outcome in COVID-19 Patients. Viruses 2023, 15, 1904. https://doi.org/10.3390/v15091904.
Akkiz, H. Unraveling the Molecular and Cellular Pathogenesis of COVID-19-Associated Liver Injury. Viruses 2023, 15, 1287. https://doi.org/10.3390/v15061287.
Allameh, A.; Niayesh-Mehr, R.; Aliarab, A.; Sebastiani, G.; Pantopoulos, K. Oxidative Stress in Liver Pathophysiology and Disease. Antioxidants 2023, 12, 1653. https://doi.org/10.3390/antiox12091653.
Point 14. The authors should better explain the future perspectives and directions of the research for preventing endothelial oxidative stress COVID-19-related.
Answer 14: Research on preventing endothelial oxidative stress in the context of COVID-19 is ongoing, and several potential future perspectives and directions can be considered, which can include: 1/ New antioxidant therapies; 2/ Mitochondrial protection; 3/Targeting therapy directed towards inflammatory pathways; 4/ Combination therapies and multidisciplinary collaboration; 5/ Oxidative stress biomarkers monitoring of COVID-19 and Post-COVID-19 patients.
1/ Investigation of antioxidants as therapeutic agents could focus on determining the optimal dosages, timing, and administration routes for these antioxidants in COVID-19 patients to mitigate endothelial dysfunction.
2/ Mitochondrial protection strategies can include maintaining or enhancing mitochondrial health in endothelial cells by drugs or interventions that protect mitochondria from oxidative damage.
3/Targeting therapy directed toward inflammatory pathways could investigate drugs or interventions given the close connection between oxidative stress and inflammation. The combination of anti-inflammatory agents in combination with antioxidants, to address endothelial dysfunction comprehensively and OS.
4/ Combination therapies and multidisciplinary collaboration directed to investigating the potential synergistic effects of different therapeutic approaches including a combination of antioxidants and antiviral or anti-inflammatory drugs can provide more comprehensive protection against endothelial oxidative stress in COVID-19. Interdisciplinary research in this area is crucial not only for mitigating the acute complications of COVID-19 but also for addressing potential long-term endothelial damage associated with the coronavirus disease.
5/ Oxidative stress biomarkers monitoring of COVID-19 and Post-COVID-19 patients. Identify reliable biomarkers that can assess endothelial oxidative stress in COVID-19 patients. Biomarkers could help in the early detection and monitoring of endothelial dysfunction or the long-term effects of COVID-19 on endothelial function and OS-related damages, which can guide therapeutic decisions. On the other side, genetic and metabolic profiling could help identify individuals who are more susceptible to critical levels of ROS and RNS and tailor interventions accordingly.

Round 2
Reviewer 1 Report
The manuscript has been significantly improved.
The English usage is fine; however, minor refinement is needed during the proofreading.
Reviewer 3 Report
The manuscript has been improved and can be published.
Editing of English language is suggested.